# TRACE-BACK ALONG CAPSULES AND ITS APPLICATION ON SEMANTIC SEGMENTATION

## ABSTRACT

In this paper, we propose a capsule-based neural network model to solve the semantic segmentation problem. By taking advantage of the extractable part-whole dependencies available in capsule layers, we derive the probabilities of the class labels for individual capsules through a recursive, layer-by-layer procedure. We model this procedure as a traceback pipeline and take it as a central piece to build an end-to-end segmentation network. Under the proposed framework, image-level class labels and object boundaries are jointly sought in an explicit manner, which poses a significant advantage over the state-of-the-art fully convolutional network (FCN) solutions. Experiments conducted on modified MNIST and neuroimages demonstrate that our model considerably enhance the segmentation performance compared to the leading FCN variant.

## 1 INTRODUCTION

An effective segmentation solution should have a well-equipped mechanism to capture both semantic (i.e., *what*) and location (i.e., *where*) information. The fully convolutional network (FCN) (Long et al., 2015) and its variants (Ronneberger et al., 2015; Noh et al., 2015; Badrinarayanan et al., 2017) constitute a popular class of solutions for this task, producing state-of-the-art results in a variety of applications. FCN and its variants (FCNs) are commonly constructed with an encoder-decoder architecture. In the encoding path, input images are processed through a number of "convolution + pooling" layers to generate high-level latent features, which are then progressively upsampled in the decoder to reconstruct the target pixel labels. The feature maps produced in higher (coarser) layers and those in lower (finer) layers contain complementary information: the former is richer in semantics, while the latter carries more spatial details that define class boundaries.

Originated from and constructed upon convolutional neural networks (CNNs) (Krizhevsky et al., 2012; Simonyan & Zisserman, 2014), FCNs' encoders inherit some common drawbacks of CNNs, one of which is the lack of an internal mechanism in achieving viewpoint-invariant recognition. Traditional CNNs, as well as FCNs, rely on convolution operations to capture various visual patterns, and utilize poolings to enable multi-scale processing of the input images. Rotation invariance, however, is not readily available in both models. As a result, more data samples or additional network setups (Cohen & Welling, 2016; Cohen et al., 2018) would be required for objects from different viewpoints to be correctly recognized. The absence of explicit part-whole relationships among objects imposes another limitation for FCNs – without such a mechanism, the rich semantic information residing in the higher layers and the precise boundary information in the lower layers can only be integrated in an implicit manner (Zhang et al., 2018).

Capsule nets (Sabour et al., 2017; Hinton et al., 2018), operating on a different paradigm, can provide a remedy. Capsule nets are built on capsules, each of which is a group of neurons representing one instance of a visual entity, i.e., an object or one of its parts (Hinton et al., 2011). Capsules output both activation probabilities of their presence and the instantiation parameters that describe their properties, such as pose, deformation and texture, relative to a viewer (Hinton et al., 2011). During inference propagation, the principle of coincidence filtering is employed to activate higher-level capsules and set up part-whole relationships among capsule entities. Such part-whole hierarchy equips capsule nets with a solid foundation for viewpoint-invariant recognition, which can be implemented through dynamic routing (Sabour et al., 2017) or EM routing (Hinton et al., 2018). The same hierarchy, if

properly embedded into a segmentation network, would provide a well-grounded platform to specify contextual constraints and enforce label consistency.

With this thought, we develop a capsule-based semantic segmentation solution in this paper. Our approach treats capsule nets as probabilistic graphical models capable of inferring probabilistic dependences among visual entities, through which part-whole relationships can be explicitly constructed. As a concrete implementation, we propose a new operation sequence, which we call *traceback pipeline*, to capture such part-whole information through a recursive procedure to derive the class memberships for individual pixels. We term our model *Tr-CapsNet*.

The contributions of our Tr-CapsNet can be summarized as:

1. In Tr-CapsNet, the class labels for individual spatial coordinates within each capsule layer are analytically derived. The traceback pipeline in our model, taking advantage of the graphical properties of capsule nets, is mathematically rigorous. To the best of our knowledge, this is the first work to explore a capsule traceback approach for image segmentation. In addition, probability maps at each capsule layer are readily available, which makes it convenient to conduct feature visualization and layer interpretation.

2. In parallel with segmentation, Tr-CapsNet carries out explicit class recognition at the same time. Such explicitness poses a powerful practical advantage over FCNs.

3. The traceback pipeline is designed under a general context, making it applicable to many other potential tasks, including object localization and detection, action localization and network interpretation.

## 2 BACKGROUND

A capsule (Sabour et al., 2017; Hinton et al., 2018) is aimed to represent an instance of a visual entity, and its outputs include two parts: 1) the probability of the existence of this instance and 2) instantiation parameters, which carries certain visual properties of the instance. Built on capsules, capsule nets are designed to overcome the drawbacks of CNNs, in particular, the inability of handling viewpoint changes and the absence of part-whole relationships among visual entities.

Unlike CNNs, where image patterns are captured through convolution (correlation) operations, capsule nets rely on high-dimensional coincidence filtering (HDCF) to detect objects that attract concentrated votes from their parts/children. The HDCF mechanism is carried out based on the computation of two network parameter sets: weights and coupling coefficients, of the connected capsules in consecutive layers (Fig. 1).

In CNNs, the weights between neurons represent convolution filters, which are to be learned in training. In capsule nets, each weight matrix $W$, however, represents a linear transformation that would map the parts of an object into a cluster for the same whole. Similar to CNN, weight matrices in capsule nets are globally learned in training through labeled samples. As viewpoint changes do not alter the relative orientation between parts and whole, maintaining the same $W$ would be sufficient to handle inputs from different angles, leading a viewpoint-invariant system. The coupling coefficients $c_{ij}$ are assignment probabilities between a pair of whole-part capsules. Different from the weights, $c_{ij}$ is dynamically determined solely in inference time. The $c_{ij}$s between each capsule $i$ and all its potential parent capsules are summed to 1, and thus have a similar effect of average pooling in CNNs.

While the two versions of capsule nets (Sabour et al., 2017; Hinton et al., 2018) follow the same HDCF general principle for object recognition, they differ in several aspects, primarily the setup of the instantiation parameter $u$ and the routing process to estimate $c_{ij}$. In Sabour et al. (2017), $u$ is set as a one-dimensional vector, whose length represents the probability of the capsule's presence. The routing algorithm, *dynamic routing*, updates $c_{ij}$ indirectly by the scalar product of votes and outputs of possible parents. Meanwhile, Hinton et al. (2018) uses a $4 \times 4$ matrix to represent instantiation parameters of a capsule, and formulates the routing procedure as a mixture of Gaussians problem, which is then solved with a modified Expectation-Maximization algorithm.

In this work, we take advantage of the part-whole relationships available in capsule nets to produce the labels for the pixels at the input image space. To facilitate the derivation of our solution, we would first summarize the related concepts under a probabilistic framework.

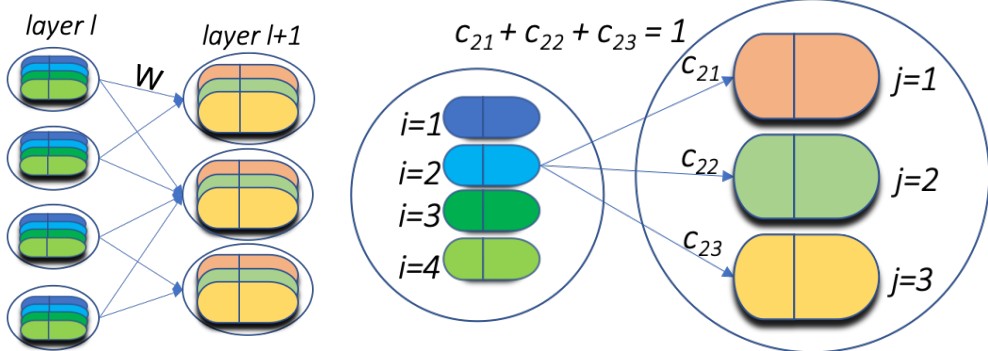

Figure 1: Left diagram: spatial relationships between two contiguous capsule layers (shown in 1D for clarity). If a position is in the receptive field of another position in the higher layer, the two positions are connected with an arrow pointing to the higher layer. Each capsule at the higher-layer position is a possible parent of capsules at its connected lower-layer position. Right diagram: the assignment probabilities and its normalization requirement. Refer to text for more details.

Capsules can be categorized into different types (*e.g.* cat, bicycle, or sofa). Each capsule layer $l$ is associated with a set of capsule types $\mathcal{T}^l = \{t_1^l, t_2^l, ..., t_m^l, ...\}$. At each spatial coordinate (position) of certain layer $l$, there exists exactly one capsule of every type in $\mathcal{T}^l$. The probability of a capsule $i$ existing/activated in the network is denoted as $P(i)$. If capsule $i$ on layer $l$ is within the receptive field of the capsule $j$ of layer $l+1$, we call $j$ a possible parent of $i$. The assignment probability of $i$ being a part of $j$ is denoted as $P(j|i) = c_{ij}$. The assignment probabilities between capsule $i$ and the capsules co-located with $j$ are forced to meet the normalization requirements of $\sum_{j \in \mathcal{T}^{l+1}} c_{ij} = 1$ (Sabour et al., 2017; Hinton et al., 2018).

Three types of capsule layers were introduced in (Sabour et al., 2017; Hinton et al., 2018), which will also be used as building blocks in our Tr-CapsNet. Therefore, we briefly describe them as follows,

- Primary capsule layer is the first capsules layer, where features from previous convolution layer are processed and transitioned into capsules via convolution filtering.

- Convolutional capsule layer(s) function similarly to CNN convolutional layers in many aspects. However, they take capsules as inputs and utilize certain routing algorithm to infer outputs, which are also capsules.

- Class capsule layer $L$ is a degenerated layer with one capsule for each predefined class label $\mathcal{C}_k \in \mathcal{C} = \{\mathcal{C}_1, \mathcal{C}_2, ..., \mathcal{C}_k, ...\}$. Each capsule in the previous layer is fully connected to the capsules in this layer.

The capsule nets are designed for object classification. In this work, we aim to take advantage of the notion and structure of capsule nets to design a semantic segmentation solution, hoping the viewpoint invariance and network interoperability mechanism can lead to improved segmentation performance. We present the description of our model in next section.

## 3 THE ARCHITECTURE OF OUR TR-CAPSNET

The general structure of our Tr-CapsNet, which consists of three components, is shown in Fig. 2. It starts with a *feature extraction module* that capture the discriminative features of the input data to be fed into the later modules. For this purpose, a sequence of convolution layers as in CNNs would be a good choice (Sabour et al., 2017; Hinton et al., 2018). A *capsule & traceback module* comes next. It consists of a sequence of capsule layers and a traceback pipeline, followed by a convolution layer. The lineup of the capsule layers, shown as orange and brown arrows in Fig. 2, can be a combination of one primary capsule layer, optionally one or more convolutional capsule layers, and a class capsule layer at the end. The traceback pipeline, shown as red arrows in Fig. 2, is the major innovation of this paper. It produces class maps of the same size as the primary capsule layer, which are then taken as

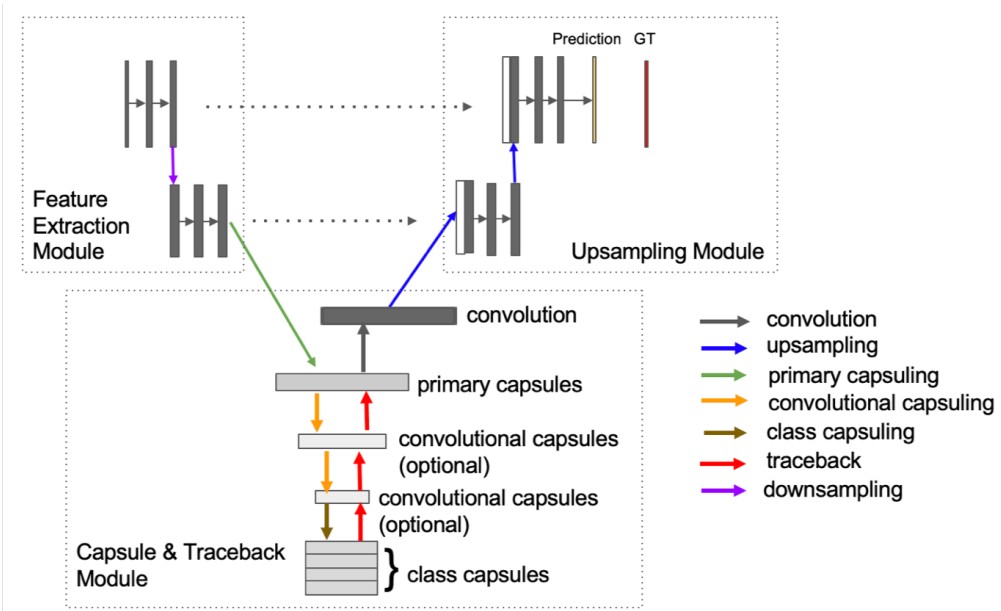

Figure 2: Overall architecture of our Tr-CapsNet. The traceback pipeline, shown as red arrows, is the major innovation of this paper. *GT* stands for ground-truth. Refer to text for details.

inputs to a convolution layer. This pipeline is specifically designed for segmentation purpose, and its details will be presented in next subsection. The third component of our model, a *upsampling module*, restores the original resolution by upsampling the label map computed in the capsule & trackback module. This module is implemented based on the deconvolution scheme in Long et al. (2015) in this paper, but it can also be implemented through atrous convolution (Chen et al., 2018), unpooling (Badrinarayanan et al., 2017), or simply bilinear interpolation. It should be noted that the feature extraction and upsampling layers can be regarded as somewhat symmetric in terms dimensionality, where the latter can be optional if no dimensionality reduction occurs in feature extraction layers.

## 3.1 TRACEBACK PIPELINE: DESIGN AND DERIVATIONS

The ultimate goal of image segmentation is to compute the probability of each pixel belonging to certain class type, hopefully in great accuracy. The traceback pipeline is designed to serve this purpose. It should be noted that, over the inference procedure of capsule nets, the probability of each capsule $P(i)$ and the assignment probabilities between contiguous layers of capsules $c_{ij}$ are calculated and become available. With that, $P(\mathcal{C}_k)$, the probability of a class label for each location in the capsule layers, can be potentially inferred through repeated applications of the product rule and the sum rule in probability theory. If this process is carried out, it would essentially trace the class labels in the class capsule layer in a backward-propagation manner, layer by layer until it reaches the first capsule layer. We name this recursive, layer-by-layer process *traceback pipeline*. Hinton et al. (2000) takes a similar approach, which interprets an image as a parse tree to perform recognition and segmentation simultaneously. Their model was constructed based on graphical models though, with no capsule or other neural network involved.

The detailed traceback procedure is explained as follows. The feature extraction layers and the capsules layers of Tr-CapsNet are adopted from the capsule nets. Therefore, same as in the latter, $P(i)$ and $c_{ij}$ are available during the inference procedure of Tr-CapsNet. With that, the probability of a position belonging to certain class $P(\mathcal{C}_k)$, in layer $l$, can be calculated as

$$P(\mathcal{C}_k) = \sum_{i \in \mathcal{T}^l} P(\mathcal{C}_k, i) = \sum_{i \in \mathcal{T}^l} P(i)P(\mathcal{C}_k|i), \qquad (1)$$

where $i$ is a capsule type associated with layer $l$ and $P(\mathcal{C}_k|i)$ shows the likelihood of certain position taking $\mathcal{C}_k$ as its class label, given that a capsule with type $i$ is located on it. With $P(i)$ of layer $l$ being available over the inference procedure, $P(\mathcal{C}_k|i)$ is the only term to be estimated.

Let $L$ be the index of the class capsule layer. The $P(\mathcal{C}_k|i)$ at its immediately previous layer, $L-1$, would be the assignment probability between the capsule $i$ of the layer $L-1$ and the class capsule $\mathcal{C}_k$ on layer $L$. Again, $P(\mathcal{C}_k|i)$ is available after inference reaches layer $L$.

The $P(\mathcal{C}_k|i)$ on other layers, however, needs to be solved. After some simple mathematical derivations, we found that the estimation of $P(\mathcal{C}_k|i)$ could be written into a recursive equation w.r.t. the upper layer, if we assume that each lower-layer capsule only takes capsules in one particular position of the higher-layer as its possible parents. Let capsule $j$ of layer $l+1$ and capsule $i$ of layer $l$ form a possible parent-child pair. Then the conditional probability $P(\mathcal{C}_k|i)$ can be computed as

$$
\begin{aligned}
P(\mathcal{C}_k|i) &= \sum_{j \in \mathcal{T}^{l+1}} P(\mathcal{C}_k, j|i) \quad (i \text{ is assigned to the parent } j.) \\
&= \sum_{j \in \mathcal{T}^{l+1}} P(j|i) P(\mathcal{C}_k|j, i) \\
&= \sum_{j \in \mathcal{T}^{l+1}} c_{ij} P(\mathcal{C}_k|j) \quad (i \text{ has the same class as its parent } j.)
\end{aligned}
\tag{2}
$$

The recurrence in Eqn. (2) indicates that the estimation of $P(\mathcal{C}_k|i)$ requires the conditional probabilities $P(\mathcal{C}_k|j)$ from the parent layer.

We should note that when the inference propagation reaches layer $L$, all $c_{ij}$ are available. In addition, the $P(\mathcal{C}_k|j)$ are also available for layer $L-1$. With this combination, the $P(\mathcal{C}_k|i)$ on other layers can be computed through a traceback process. More specially, $P(\mathcal{C}_k|i)$ can be estimated with a layer-by-layer backward propagation procedure – starting at the layer $L-1$, and repeatedly applying Eqn. (2) to compute the conditional probabilities for the lower layers. We use a term *traceback-depth* to indicate the number of capsule layers to be traced along the pipeline. The size of the last capsule layer, of which the conditional probabilities are calculated, is termed *traceback-size*.

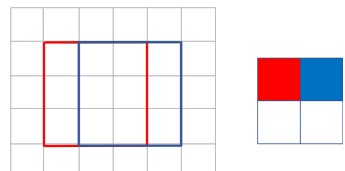

Figure 3: Illustration of a *parents-at-multi-positions* scenario. Left: a capsule layer, where each cell represents a capsule. Right: four capsules in the next layer, generated from the previous layer (left) through a 3×3 convolution with stride 1. The capsules in the overlapping area of the two frames (left) take both the red and blue capsules in the higher-level layer (right) as parents.

Eqn. (2) is for a simple case where each lower-layer capsule only takes same-position capsules of the higher-layer as its possible parents. For the convolutional capsule layers, however, capsules in two or more positions might be the parents of a lower-layer capsule. Fig. 3 shows an example where a number of capsules take capsules at different positions in next layer (blue and red) as their parents. For these cases, the traceback procedure remains effective, but the computation of $P(\mathcal{C}_k|i)$ needs to be modified. One possible approach is to stipulate $P(\mathcal{C}_k|i) = \sum_n P_n(\mathcal{C}_k|i)/N$, where $n$ represents $n$-th location for possible parents of the capsule $i$ and $N$ is total location number. This summation bears some resemblance to the summation in deconvolution operation (Long et al., 2015) that adds two overlapping filter responses.

## 3.2 Loss Function

The loss function of our model, $L = \lambda_1 L_{\text{margin}} + \lambda_2 L_{\text{ce}}$, is a weighted sum of a margin loss term $L_{\text{margin}}$ over the class capsules and a pixel-wise softmax cross-entropy $L_{\text{ce}}$ over the final feature map. $\lambda_1$ and $\lambda_2$ are the weighting parameters.

The margin loss, adopted from (Sabour et al., 2017), is for recognition, which strives to increase the probabilities of class capsules for the corresponding objects existing in the input image. Let $P(\mathcal{C}_k)$ denote the probability of the class capsule k. For each predefined class, there is a contributing term

$L_k$ in the margin loss

$$L_k = T_k \, max(0, m^+ - P(\mathcal{C}_k))^2 + \lambda_{\text{margin}}(1 - T_k) \, max(0, P(\mathcal{C}_k) - m^-)^2 \qquad (3)$$

where $m^+ = 0.9$, $m^- = 0.1$, and $\lambda_{\text{margin}} = 0.5$. $T_k = 1$ when a class $k$ is present in the mask and $T_k = 0$ otherwise. The margin loss is the summation of all $L_k$, i.e. $L_{\text{margin}} = \sum_k L_k$.

The margin loss plays a key role in training the model because it is the driving force for the initiation and convergence of the capsule layers in the model. While the cross-entropy loss handles the boundary localization in semantic segmentation, the margin loss is responsible for the object recognition. As a result, the ratio $\frac{\lambda_1}{\lambda_2}$ in the total loss equation is an important hyper-parameter to tune the performance of our models, which we will discuss in next section.

## 4 EXPERIMENTAL RESULTS

**Datasets**   The effectiveness of our model is evaluated with two datasets: modified MNIST and Hippocampus dataset from the Alzheimer's Disease Neuroimaging Initiative (ADNI) project.

The MNIST (LeCun & Cortes, 1998) dataset contains 28×28 images of handwritten digits. In our experiments, random noise is added to each image to broaden the intensity spectrum and increase the numerical stability. Ground-truth segmentations are generated by filtering original foreground digit pixels of each image with an intensity threshold. Two experiments are conducted on this noise-added MNIST dataset. The first one is a multi-digit segmentation test, in which competing methods are evaluated based on their performance in segmenting all ten digits. The second experiment is a robustness test to assess the models under various occlusion scenarios.

The second dataset contains brain images and the corresponding Hippocampus masks of 110 subjects, both downloaded from ADNI website (adni.loni.usc.edu). The images to be segmented are T1-weighted whole brain Magnetic resonance (MR) images (Fig. 4, left), and the ground-truth segmentations were generated through semi-manual boundary delineation by human raters. In order to alleviate the data imbalance problem, we roughly aligned all brain MRIs, and cropped them with a common region that encloses the right hippocampi of all subjects, which led to subvolumes of size 24×56×48.

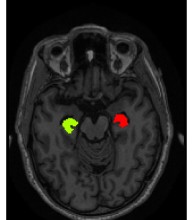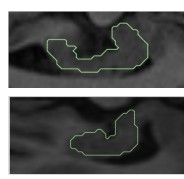

Figure 4: Left: Hippocampi in a brain MR scan. Right: zoom-in view of two cropped slices, superimposed with contours of the ground-truth Hippocampus masks.

Two-dimensional slices, sized 24×56, along the axial view of all the three-dimensional volumes are taken as inputs in our experiments (Fig. 4, right). The resulted binary segmentations of each subject are then stacked to constitute the final 3D segmentations for evaluation. Ten-fold cross-validation is used to train and evaluate the performance of the competing models.

As partially observable from Fig. 4, human Hippocampi have rather complicated 3D shapes, surrounded by tissues of similar intensity profiles. The image contrasts in many boundary areas are very vague or even nonexistent, which makes accurate and reproducible delineations very difficult, even for human experts. In addition, the Hippocampus masks in ADNI were generated with a 3D surface fitting solution on 42 salient surface points identified by human raters. This fitting procedure inevitably brings noise to the ground-truth, imposing extra difficulty for the segmentation task. Tong et al. (2013) (based on sparse coding & dictionary learning), Song et al. (2015); Wu et al. (2015) (both based on sparse coding) and Chen et al. (2017) (based on multiview U-Net) are some of the recently published Hippocampus segmentation works that also used ADNI data. It should be noted that direct comparisons of the solutions are often not feasible, as different datasets, subjects and ground-truth setups have been used in the studies.

**Evaluation metrics**   Totally four metrics are adopted for performance evaluations. *Pixel accuracy* (PA), *mean class accuracy* (MCA) and *Dice ratio* are calculated in our experiments to evaluate the segmentation accuracies in a general sense. In the occlusion tests, we use an additional metric, number of *added holes* (AH), to assess the segmentation consistency of each model, in terms of

topology preservation. An added hole is a connected, wrongly labeled area that completely resides within a correctly labeled segment. The AH (or more accurately, the absence of AH) provides a discrete metric to measure the segmentation consistency within a whole object.

**Baselines** Modified U-Nets (Ronneberger et al., 2015) are used as the baseline models for comparison. In the MNIST experiment, the U-Net was altered to have 5 convolution layers along the encoder path and 5 deconvolution layers along the decoder path. Pooling layers are discarded, and we replace them with strides to reduce the feature-map dimensions along the encoding path. In the Hippocampus experiment, padding is added to the original U-Net to maintain the spatial dimension of the output of each convolution/deconvolution layer (Table 5 in the Appendix A). We only keep one convolutional layers prior to each pooling layer. Explicit data augmentation step in U-Net is replaced with dropout (Srivastava et al., 2014) to reduce overfitting. To explore the effect of receptive fields sizes, we implement four modified U-Nets with different pooling schemes to assign the bottom layers with various scales of feature maps. In reporting the performance of different versions, we add a digit at the end of U-Net to indicate the dimension of the smallest feature map (e.g., U-Net-6 means the dimension of the feature map at the coarsest layer is 6×6).

**Implementation details** For the 10-digit experiment, we take a regular convolution layer as the feature extraction layer for our Tr-CapsNet. We feed 24 types of capsules associated with the following primary capsule layer and each capsule outputs a 8D vector. The primary capsule layer is followed by the class capsule layer. The traceback pipeline is applied between the class capsule layer and primary capsule layer. We trained three versions of Tr-CapsNet, with different sizes of label maps (or dimensions of the positions) maintained in the primary capsule layer. More specially, we use 7×7, 9×9 and 11×11 positions, and the corresponding models are named as Tr-CapsNet-7, Tr-CapsNet-9 and Tr-CapsNet-11.

In the Hippocampus experiment, the feature extraction layers are adopted from first layers of the encoder in the baseline U-Net models. For the capsule & traceback module, traceback-depth 1 and 2 are implemented. In the models with traceback depth 1, there are 32 capsule types in the primary capsule layers, outputting 8D vectors. The class capsule layer in these models has one 16D vector for the 2 class capsules (hippocampus and background). The traceback pipeline is between the class capsule layer and primary capsule layer. In the model with traceback-depth 2, one convolution capsule layer is inserted between the primary capsule layer and the class capsule layer. For this model, we set 64 capsules types in the primary layers with 3D vectors, 24 capsule types in the convolution layer with 8D vectors, and followed by the class capsule layer. The traceback pipeline in this model is along these three capsule layers.

In Tr-Capsnet, the ratio of the two hyper-parameters ($\frac{\lambda_1}{\lambda_2}$) in the loss function need to be tuned. In the MNIST experiment, the ratio is selected from the range of $\{1 \sim 5\}$ with an interval of 1. In the Hippocampus experiment, we observed that the recognition task was far more difficult, largely due to the greater complexity of the dataset. Accordingly, we selected the ratio from a larger range, $\{1 \sim 20\}$ with an interval of 1. A large weight on recognition term is presumed to force the network to build more accurate part-whole relationships, which lays a solid foundation to produce more accurate overall segmentations.

The dynamic routing algorithm in (Sabour et al., 2017) is adopted in all experiments to estimate coupling coefficients. All networks are implemented in TensorFlow (Abadi et al., 2016) and trained with the Adam optimizer (Kingma & Ba, 2014). All experiments are run on a single Nvidia GTX TITIAN X GPU with 12GB memory.

## 4.1 RESULTS ON MODIFIED MNIST

In this experiment, we tried different weight ratios between margin loss and cross-entropy to evaluate the importance of the individual loss components. We also explored setups of U-Nets with different receptive scales. The results are summarized in Table 1. The number of parameters for each model is listed. Among the baseline U-Nets, U-Net-6 (6 stands for the dimension of the smallest feature map at the bottom layer) gets the best results, so we only include it in the table. From the table, it is evident that our Tr-CapsNet outperform the best U-Net model in all metrics for this dataset. We can also tell that two factors, dimension of primary capsule layer (7/9/11) and weights for loss terms, affect the performance of our Tr-CapsNet.

| Method | Loss Weights $(\lambda_1, \lambda_2)$ | Number of Training Parameters | PA | Mean Accuracy | Dice Ratio |
|---|---|---|---|---|---|
| Tr-CapsNet-7 | 1, 1 | 5.09M | **99.06** | **99.07** | $99.19 \pm 5.81$ |
| Tr-CapsNet-9 | 2, 1 | 3.40M | 99.01 | 99.02 | $99.23 \pm 5.22$ |
| Tr-CapsNet-9 | 1, 1 | 3.40M | 99.04 | 99.05 | $\mathbf{99.27 \pm 5.05}$ |
| Tr-CapsNet-11 | 1, 1 | 2.24M | 98.31 | 98.31 | $98.89 \pm 5.34$ |
| U-Net-6 | - | 3.14M | 98.04 | 98.03 | $95.63 \pm 6.13$ |

Table 1: Results on the 10-digit experiment. Refer to text for details.

## 4.2 RESULTS ON HIPPOCAMPUS DATASET

**Effects of size of the primary capsule layer** In order to explore how the dimension of primary layer (number of positions) plays a role in the segmentation procedure, we train different versions of Tr-CapsNet, as we do for MNIST experiments. The results on one particular split of the cross-validation test is shown in Table 2. In column 3, we list the traceback-size and traceback-depth of each model. From the results, it appears that the size of primary layer size and the weights of the loss terms both affect the results to certain extent.

For our Tr-CapsNet, when the primary capsule size set to $4\times20$ and loss weights set to 15:1, we obtain the most accurate segmentation results. Reducing the dimension of primary layer, as well as decreasing the contribution of margin-loss, seems worsening the model performance. One possible explanation is that when the the dimension of primary layer is set to a very small number, the positions of the primary capsules may not be precise enough to represent the visual entities at the input image level.

For the contribution of margin loss increasing it should make the network strive to obtain more accurate overall recognition. With our mathematically rigorous traceback, higher recognition accuracy would translate into more precise membership labelling at individual pixels.

For U-Nets, with a fixed input size, the feature map size of the bottom layer is the reciprocal of the size of the receptive field of this layer. Setting the feature map size to a small number would allow the information of pixels from a broader range to be integrated, leading to improved class decision and boundary localization. This trend can be spotted in Table 2 from U-Net-$3\times7$ to U-Net-$4\times20$.

| Method | Number of Training Parameters | Traceback-size and Depth | Loss Weights $(\lambda_1, \lambda_2)$ | Dice Ratio |
|---|---|---|---|---|
| Tr-CapsNet | 1.50M | $2\times6$, 1 | (7, 1) | $87.51 \pm 2.378$ |
| Tr-CapsNet | 1.57M | $4\times12$, 1 | (10, 1) | $88.05 \pm 2.541$ |
| Tr-CapsNet | 1.90M | $4\times12$, 2 | (10, 1) | $88.19 \pm 1.874$ |
| Tr-CapsNet | 3.02M | $4\times20$, 1 | (15, 1) | $\mathbf{88.86 \pm 1.628}$ |
| Method | Number of Training Parameters | Smallest Feature Map | - | Dice Ratio |
| U-Net | 1.14M | $3\times7$ | - | $\mathbf{88.41 \pm 1.707}$ |
| U-Net | 0.95M | $4\times12$ | - | $88.27 \pm 1.778$ |
| U-Net | 2.19M | $4\times20$ | - | $87.68 \pm 2.219$ |

Table 2: Effect of the traceback pipeline. Refer to text for details.

**Overall average performance** Through the one-split validation, we identified the potentially best setups for both Tr-CapsNet and modified U-Net, which are Tr-CapsNet- $4\times20$ and U-Net-$3\times7$, respectively (refer to APPENDIX A for detailed layer configurations). We then carried out a ten-fold cross-validation on the entire 110 data points. The average Dice ratio for Tr-CapsNet- $4\times20$ is 87.25 with a standard deviation 5.05, while U-Net-$3\times7$ obtains 86.23 with a standard deviation 2.19. In other words, our model outperforms the best U-Net model.

### 4.3 OCCLUSION TEST

| Method | Loss Weights ($\lambda_1$, $\lambda_2$) | PA | Dice | AH | 0 | | 8 | |
|---|---|---|---|---|---|---|---|---|
| | | | | | Dice | AH | Dice | AH |
| Tr-CapsNet-9 | (2, 1) | **93.49** | **95.44** | **264** | 93.89 | 185 | **96.99** | **79** |
| Tr-CapsNet-9 | (3, 1) | 92.73 | 95.05 | 270 | **95.36** | **123** | 94.74 | 147 |
| U-Net-6 | - | 91.17 | 91.20 | 300 | 92.59 | 123 | 89.82 | 177 |

Table 3: Results on occluded test. Refer to text for details.

One of the fundamental differences between our Tr-CapsNet model and FCNs lies in the fact that Tr-CapsNet does segmentation and recognition simultaneously. The recognition subtask, if conducted well in practice, would equip Tr-CapsNet with an additional power in dealing with adversarial situations, which include input images with occlusions. In order to evaluate and demonstrate the effect, we design an occlusion test as follows.

We train our model on images of two easily confused digits (e.g. 0 and 8) with the modified MNIST samples. For the test samples, however, we generate occlusions in each image by setting the intensities of several horizontal lines around the image centers to black. The segmentation results are shown in Table 3. It is evident that Tr-CapsNet achieve significantly higher accuracies than the best baseline model.

Table 4 shows several representative prediction images from the occlusion test. Pixels classified into digit 8 are shown in green color and red color pixels have been classified as digit 0. In the first column, U-Net-6 generates rather precise boundary localization but makes totally wrong labellings for all pixels. Our Tr-CapsNet gets both aspects, overall classification and boundary localization, very well worked out. In column 2, U-Net-6 makes the same mistake for the entire digit, while Tr-CapsNet put correct labels for vast majority of the pixels. U-Net-6 performs correctly for the input in column 3. Overall, Tr-CapsNet appears to be able to generate more robust and consistent segmentation results, and this capability should be attributed to the recognition component and the part-whole mechanism built in our model.

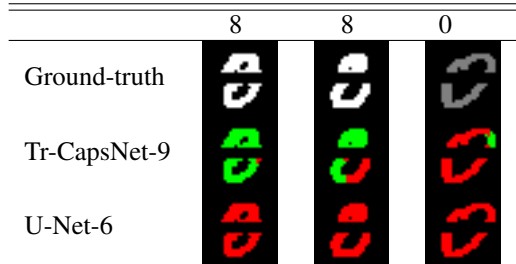

Table 4: Prediction results from the occlusion test. Refer to text for details.

## 5 DISCUSSION AND RELATED WORK

Since the inception of the first FCN model, FCN variants have become the most popular solutions for semantic segmentation. Compared with traditional non-deep learning techniques as well as patch-based CNN solutions, the efficacy of FCNs should be attributed, in large part, to their capability to process information from various spatial scales.

**FCN models** While powerful, FCNs also have some inherent limitations. Originated from CNNs, FCNs tend to lose precise spatial information along the pooling operations/layers, and inconsistent pixel labelings can be resulted from the limited receptive field at each neuron. A number of approaches have explored to add the contextual information back to the CNN component. Conditional Random Fields (CRFs) based solutions (Chen et al., 2014; 2018) refine the final segmentation results with a post-processing stage to boost its ability to capture fine-grained details. Dilated convolutions (Yu & Koltun, 2015; Yu et al., 2017) are utilized to take advantage of the fact that the receptive fields of systematic dilation can be expanded exponentially without losing resolution. Fusion of global features extracted at lower layers and local features from higher layers through skip connections have

also been well studied (Long et al., 2015; Ronneberger et al., 2015). To produce segmentation results with certain shape constraints, (Ravishankar et al., 2017; Chen et al., 2013) integrate shape priors into existing deep learning framework with shape-based loss terms. New models have emerged in the past 18 months or so, such as multi-resolution fusion by RefineNet (Lin et al., 2017), pyramid pooling module in PSPNet (Zhao et al., 2017), Large kernel matters (Peng et al., 2017), Clusternet (LaLonde et al., 2018), atrous convolution and fully connected CRFs (Zhang et al., 2018), joint weakly-supervised and semi-supervised learnings (Wei et al., 2018) and Y-Net (Mehta et al., 2018).

**Capsules, capsule nets and capsule-based segmentation solutions** The history of the notion of capsules can be dated back to (Hinton, 1981), in which the author proposed to describe an object over the viewpoint-invariant spatial relationships between the object and its parts. This model should be regarded as the first theoretical prototype of the latest capsule nets (Sabour et al., 2017; Hinton et al., 2018). (Zemel et al., 1990) added probability components into this prototype and encoded both probabilities and viewpoint-invariance in a fully-connected neural network. The instantiation parameters of Zemel's model were hand-crafted from input data.

The issue of how to initialize instantiation parameters was partially addressed in (Hinton et al., 2011), but their model requires transformation matrices to be input externally. Regarding the routing algorithms, (Zemel et al., 1990) introduced the notion that an object can be activated by combining the predictions from its parts, which was materialized as an averaging operation in (Hinton et al., 2011). Sabour et al. (2017); Hinton et al. (2018) took a step forward and resorted to routing-by-agreement algorithms to simultaneously activate objects and set up part-whole relationships.

To the best our knowledge, the first capsule-based semantic segmentation model has been proposed by (LaLonde & Bagci, 2018). The authors made two modifications to the original dynamic routing algorithm, such that the number of parameters is greatly reduced, which enables the model to operate on large images sizes. Deconvolutional capsule layers were devised and appended directly to convolutional capsule layers to restore the original resolution. Similar to FCNs, this model utilizes skip connections to integrate contextual information from different spatial scales. Our Tr-CapsNet, however, is designed under a different paradigm. Taking the extractable part-whole information as the foundation, label assignments in Tr-CapsNet are explicit, interpretable, and mathematically well-grounded. The decoder of our model is much simpler than that of (LaLonde & Bagci, 2018) model, with no trainable parameters (up to deconvolution layers) and no capsule-wise routing needed. Because of the analytical derivations, our model has great potential to be applied to many other applications, including object localization, detection and visualization of heap maps.

**From capsule-based recognition to capsule-based segmentation and beyond** Both versions of capsule nets have an assumption in their constructions: "at each location in the image, there is at most one instance of the type of entity that a capsule represents." (Sabour et al., 2017). Essentially, capsule nets, including the internal part-whole relationships, are built on concrete, physical object instances. With this assumption, capsule nets perform well when there is at most one instance of a category in the image, such as on MNIST and smallNORB (LeCun et al., 2004). For the image data that have multiple instances of same classes, capsule nets have no guarantee to outperform CNNs in recognition accuracy.

The traceback pipeline in our Tr-CapsNet does not rely on the this one-instance assumption. However, the current version Tr-CapsNet is designed based on the capsule nets, where the margin loss is specified with outputs of the class capsule layer. The inability of capsule nets in handling multi-instance-same-class cases also limits the capacity of our segmentation model, for certain type of dataset. Regarding the remedies, we believe efforts can be push forward along three fronts. The first one would rely on the continued development of capsule and capsule nets to overcome their limitations. The second possibility would be designing traceback solutions that can circumvent the multi-instance constraint. For example, if the class capsule layer can somehow be removed from the traceback pipeline, potentially we could accommodate multiple instances of the same category, starting at a convolutional capsule layer. The third group of remedies could be developed along pre-processing direction. For example, object detection can be carried out first, followed by Tr-CapsNet to generate accuracy segmentations within individual regions.

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

## APPENDIX A. LAYER CONFIGURATIONS

| Layer Name | Configuration | Output Size |
|---|---|---|
| input | – | 24×56×1 |
| conv1 | kernel:3×3, stride:1, padding:1 | 24×56×32 |
| pool1 | kernel:2×2, stride:1, padding:0 | 12×28×32 |
| conv2 | kernel:3×3, stride:1, padding:1 | 12×28×64 |
| pool2 | kernel:2×2, stride:1, padding:0 | 6×14×64 |
| conv3 | kernel:3×3, stride:1, padding:1 | 6×14×128 |
| pool3 | kernel:2×2, stride:1, padding:0 | 3×7×128 |
| pool3_dropout | dropout rate: 0.5 | 3×7×128 |
| deconv1 | kernel:2×2, stride:2, padding:0 | 6×14×128 |
| deconv1_conv | kernel:3×3, stride:1, padding:1 | 6×14×128 |
| concat1 | concatenation of deconv1 with pool2 | 6×14×192 |
| concat1_dropout | dropout rate: 0.5 | 6×14×192 |
| concat1_conv | kernel:3×3, stride:1, padding:1 | 6×14×128 |
| deconv2 | kernel:2×2, stride:2, padding:0 | 12×28×128 |
| deconv2_conv | kernel:3×3, stride:1, padding:1 | 12×28×128 |
| concat2 | concatenation of deconv2 with pool1 | 12×28×160 |
| concat2_dropout | dropout rate: 0.5 | 12×28×160 |
| deconv2_conv | kernel:3×3, stride:1, padding:1 | 12×28×128 |
| deconv3 | kernel:2×2, stride:2, padding:0 | 24×56×128 |
| deconv3_conv | kernel:3×3, stride:1, padding:1 | 24×56×128 |
| class_conv | kernel:3×3, stride:1, padding:1 | 24×56×2 |

Table 5: Layer configuration of the baseline U-Net model with its smallest feature map 3×7.

| Layer Name | Configuration | Output Size |
|---|---|---|
| input | – | 24×56×1 |
| conv1 | kernel:3×3, stride:1, padding:1 | 24×56×32 |
| pool1 | kernel:2×2, stride:1, padding:0 | 12×28×32 |
| conv2 | kernel:3×5, stride:1, padding:0 | 8×24×64 |
| conv2_dropout | dropout rate: 0.5 | 8×24×64 |
| primary capsules | modified convolution layer kernel:5×5, stride:1, padding:0, capsules:32, capsule size:8 | 4×20×(32×8) |
| class capsules | capsule layers with 1×1 kernel, capsules:2, capsule size:16 | 2×16 |
| traceback | traceback the probability map | 4×20×2 |
| conv3 | kernel:3×3, stride:1, padding:1 | 4×20×256 |
| deconv1 | kernel:5×5, stride:1, padding:0 | 8×24×128 |
| concat1 | concatenation of deconv1 with conv2 | 8×24×192 |
| concat1_dropout | dropout rate: 0.5 | 8×24×192 |
| deconv1_conv | kernel:3×3, stride:1, padding:1 | 8×24×128 |
| deconv2 | kernel:5×5, stride:1, padding:0 | 12×28×128 |
| concat2 | concatenation of deconv2 with pool1 | 12×28×160 |
| deconv2_conv | kernel:3×3, stride:1, padding:1 | 12×28×128 |
| deconv3 | kernel:4×4, stride:2, padding:0 | 26×58×128 |
| class_conv | kernel:3×3, stride:1, padding:1 | 24×56×2 |

Table 6: Layer configuration of Tr-CapsNet with its traceback size 4×20.

