# OpenReview forum: "Trace-back along capsules and its application on semantic segmentation  		"
_ICLR.cc/2019/Conference_

### Official Review · AnonReviewer3 · 2018-10-29
**Good paper which seems technically correct. Not sure if the method will generalize well beyond MNIST.**

**Rating:** 5
**Confidence:** 4

**Review:**

Based on the CapsNet concept of Sabour the authors proposed a trace-back method to perform a semantic segmentation in parallel to classification. The method is evaluate on MNIST and the Hippocampus dataset.

The paper is well-written and well-explained. Nevertheless, I think it would be useful to have some illustrations about the network architecture. Some stuff which is explained in text could be easily visualized in a flow chart. For example, the baseline architecture and your Tr-CapsNet could be easily explained via a flow chart. With the text only, it is hard to follow. Please think about some plots in the final version or in the appendix. One question which is aligned to that: How many convolutional filters are used in the baseline model?

Additionally, think about a pseudo-code for improved understandability.

Some minor concerns/ notes to the authors:
1.	At page 5: You mentioned that the parameters lambda1 and lambda 2 are important hyper-parameters to tune. But in the results you are not explaining how the parameters were tuned. So my question is: How do you tuned the parameters? In which range do you varied the parameters?
2.	Page 6; baseline model: Why do you removed the pooling layers?
3.	I’m curious about the number of parameters in each model. To have a valid discussion about your model is better than the U-Net-6 architecture, I would take into account the number of parameters. In case that your model is noticeably greater, it could be that your increased performance is just due to more parameters. As long as your discussion is without the number of parameters I’m not convinced that your model is better. A comparison between models should be always fair if two models are architectural similar.
4.	Why is the magnitude of lambda1 so different between the two dataset that you used?
5.	Could you add the inference times to your tables and discuss that in addition?
6.	What kind of noise is added to MNIST?
7.	What is the state-of-the-art performance on the Hippocampus dataset?
8.	What would be the performance in your experiments with a MaskRCNN segmentation network?
9.	I’m not familiar with the Hippocampus dataset. I missed a reference where the data is available or some explaining illustrations.
10.	For both datasets, more illustrations about the segmentation performance would be fine to evaluate your method. At least in the appendix…

My major concern is that both datasets are not dealing with real background noise. I’m concerned that the results are not transferable to other datasets and that the method shines promising just because of the simple datasets only. For example, due to the black background MNIST digits are well separated (if we skip that you added some kind of noise). So, from that point of view your results are not convincing and the discussion of your results appearing sparse and not complete.
To make your results transparent you could think about to publish the code somewhere.

---

> ### Author Response · Authors · 2018-11-24
> **Thanks & our response (1/2)**
>
> >> The paper is well-written and well-explained. Nevertheless, I think it would be useful to have some illustrations about the network architecture. Some stuff which is explained in text could be easily visualized in a flow chart. For example, the baseline architecture and your Tr-CapsNet could be easily explained via a flow chart. With the text only, it is hard to follow.
>
> Response: thanks for the assessment, and we fully agree that more illustrations should be added to make our approach more understable. In this revised manuscript, we added  figure 2 (flow chart) and rewrote the first paragraph of Architecture section to describe the modules of our Tr-CapsNet.
>
> >> Please think about some plots in the final version or in the appendix. One question which is aligned to that: How many convolutional filters are used in the baseline model?
>
> Response: we added an Appendix in this version to provide the detailed network configurations. The number of layers and number of filters in each layer are all listed there.
>
> >> 1. At page 5: You mentioned that the parameters lambda1 and lambda 2 are important hyper-parameters to tune. But in the results you are not explaining how the parameters were tuned. So my question is: How do you tuned the parameters? In which range do you varied the parameters?
>
> Response: we agree with the reviewer, and we added a paragraph on page 7 to provide more “Implementation details”.  As for the setting of lambda1 and lambda 2, “the ratio was selected from {1 ~ 5} for the MNIST experiment. In the Hippocampus experiment, we observed that the recognition task was far more difficult, largely due to the greater complexity of the dataset. Accordingly, we selected the ratio from a larger range, {1 ∼ 20} with an interval of 1.”
>
> >> 2.  Page 6; baseline model: Why do you removed the pooling layers?
>
> Response: We didn’t use max-pooling operation in the modified U-Net, based on two observations/considerations: 1) based on [1] max-pooling can simply be replaced by a convolution layer with increased stride without loss in accuracy; 2) our Tr-CapsNet doesn’t have pooling operations, and instead relies on stride to achieve multi-scale processing. We figure using a modified U-Net with wide stride instead of pooling would facilitate the comparison, analysis and future development the models.
> [1] Springenberg, Jost Tobias, Alexey Dosovitskiy, Thomas Brox, and Martin Riedmiller. "Striving for simplicity: The all convolutional net." arXiv preprint arXiv:1412.6806 (2014).
>
> >> 3. I’m curious about the number of parameters in each model. To have a valid discussion about your model is better than the U-Net-6 architecture, I would take into account the number of parameters. In case that your model is noticeably greater, it could be that your increased performance is just due to more parameters. As long as your discussion is without the number of parameters I’m not convinced that your model is better. A comparison between models should be always fair if two models are architectural similar.
>
> Response: again we fully agree with the reviewer on this regard.  In the revised version, we added the number of parameters for each model, which can be seen in Tables 1 and 2.  Two observations: 1) both U-Net and our Tr-CapsNet have variable numbers of parameters, depending on the setting, and it’s not the case that Tr-CapsNets have a lot of more parameters than U-Nets; 2) more parameters do not automatically translate into better performance. This can be seen in table 1, where Tr-CapsNet-9 (1:1) has the best performance, while it doesn’t have the most parameters. In table 2, U-Net of 1.14M parameters actually outperforms the version of 2.19M parameters.
>
> >> 4.	Why is the magnitude of lambda1 so different between the two dataset that you used?
>
> Response: As the Hippocampus dataset is obviously more complicated than MNIST, in terms of intensity separability, a large weight on recognition term is needed, and it should help to “force the network to build more accurate part-whole relationships, which lays a solid foundation to produce more accurate overall segmentations”. We added the above explanation to the revised paper.
>
> >> 5.	Could you add the inference times to your tables and discuss that in addition?
>
> Response: In both capsule nets and our Tr-CapsNets, an iterative routing-by-agreement mechanism is used. In this mechanism each capsule chooses its parent capsule in the higher layer through an iterative routing procedure. This is certainly a downside comparing with CNNs. The average inference time for one Hippocampal slice, the U-Net (feature map 4 x 20) and Tr-CapsNet (4 x 20, 1) take roughly 0.3 ms and 0.6 ms, respectively.

---

> > ### Author Response · Authors · 2018-11-24
> > **Thanks & our response (2/2)**
> >
> > >> 6.	What kind of noise is added to MNIST?
> >
> > Response: It is random noise with uniform distribution in [1-5]. The purpose of adding such noise is to broaden the intensity spectrum and increase the numerical stability.
> >
> > >> 7.	What is the state-of-the-art performance on the Hippocampus dataset?
> >
> > Response: Tong et al. (2013) (based on sparse coding & dictionary learning), Song et al. (2015); Wu et al. (2015) (both based on sparse coding) and Chen et al. (2017) (based on multiview U-Net) are some of the recently published Hippocampus segmentation works that also used ADNI data. It should be noted that direct comparisons of the solutions are often not feasible, as different datasets, subjects and ground-truth setups have been used in the studies. To provide more details about the Hippocampus segmentation data and application, we added the above sentences to the revised paper.
> >
> > >> 8.	What would be the performance in your experiments with a MaskRCNN segmentation network?
> >
> > Response: Mask R-CNN is built upon Faster R-CNN to conduct instance segmentation. For Mask R-CNN to be used for our data/applications, we may have to get detection ground-truth (bounding boxes) and train a new Faster R-CNN first.
> >
> > >> 9.	I’m not familiar with the Hippocampus dataset. I missed a reference where the data is available or some explaining illustrations.
> >
> > Response: The data (brain MRIs and Hippocampus masks) were downloaded from ADNI website: adni.loni.usc.edu. We added a paragraph in the revision to explain the data & application, with the link included.
> >
> > >> 10.	For both datasets, more illustrations about the segmentation performance would be fine to evaluate your
> >
> > Response: we agree with the reviewer. More illustrations of the segmentation results will be added in revision 2, to be uploaded early next week (Nov. 26 - 30).
> >
> > >> My major concern is that both datasets are not dealing with real background noise. I’m concerned that the results are not transferable to other datasets and that the method shines promising just because of the simple datasets only. For example, due to the black background MNIST digits are well separated (if we skip that you added some kind of noise). So, from that point of view your results are not convincing and the discussion of your results appearing sparse and not complete.
> >
> > Response: we agree with the reviewer’s assessment regarding the modified MNIST dataset. However, the Hippocampus data are rather complicated. To illustrate the dataset, we added 3 pictures, an input MR image and the ground-truth masks, in Figure 4. The complex nature of the dataset is explained as follows, which is also added to the revised paper.
> >
> > “As partially observable from Fig. 4, human Hippocampi have rather complicated 3D shapes, surrounded by tissues of similar intensity profiles. The image contrasts in many boundary areas are very vague or even nonexistent, which makes accurate and reproducible delineations very difficult, even for human experts. In addition, the Hippocampus masks in ADNI were generated with a 3D surface fitting solution on 42 salient surface points identified by human raters. This fitting procedure inevitably brings noise to the ground-truth, imposing extra difficulty for the segmentation task.”
> >
> > >> To make your results transparent you could think about to publish the code somewhere.
> >
> > Response: will certainly do. Thanks for your valuable comments.

---

> > > ### Comment · AnonReviewer3 · 2018-11-30
> > > **Thanks for your comments and your current update**
> > >
> > > Thanks for your work to improve the quality of your contribution. I'm noticing that you changed a lot and the current version of your paper addresses a lot of my concerns positively. But still, I keep my current voting for your contribution. Let me explain why:
> > >
> > > First of all, the additional segmentation results are not online which makes it still hard for me to evaluate your contribution.
> > >
> > > Second, your Table 2 confuses me. Since you are providing the results including the standard deviation, I assume that these are the results of a cross-validation. If this true, then your Table doesn't fit to your results in the text (see below the Table). Please correct me if I'm wrong. Moreover, a U-Net and your architecture is running at the same accuracy level, which makes me curious: what is now the benefit of your model? It could be clarified if you would present some images to compare the segmentation result of the U-Net and your network. Even if the performances are on the same level maybe the segmentation results are showing differences.
> > >
> > > Finally, you trained your MNIST model with uniform noise in the ratio [1-5]. Implicitly, I assume that you apply this noise level on the real image space uint8. I thought that you applied the noise to make the segmentation more challenging. But with this small noise level, even the disturbed images are easily segmentable. Hence, I don't wanna use these results to make any judgement about your method. MNIST in this setting is to easy as segmentation task.

---

> > > > ### Author Response · Authors · 2018-12-05
> > > > **Revisions & thoughts**
> > > >
> > > > >> Comment: I'm noticing that you changed a lot and the current version of your paper addresses a lot of my concerns positively.
> > > >
> > > > Response: glad to know and thanks for this comment.
> > > >
> > > > >> First of all, the additional segmentation results are not online which makes it still hard for me to evaluate your contribution.
> > > >
> > > > Response: we’re not entirely sure about what “additional segmentation results” refer to. We assume the reviewer refers to an illustration of the segmentation, which we just uploaded one in Appendix B (sorry, just realized we couldn't upload new PDF to the system). It should be noted that there may not be significant visible differences between Tr-CapsNet and U-Net results. The improvements is mainly made in a global scale, reflected in the numerical numbers.
> > > >
> > > > >> Second, your Table 2 confuses me. Since you are providing the results including the standard deviation, I assume that these are the results of a cross-validation. If this true, then your Table doesn't fit to your results in the text (see below the Table). Please correct me if I'm wrong. Moreover, a U-Net and your architecture is running at the same accuracy level, which makes me curious: what is now the benefit of your model? It could be clarified if you would present some images to compare the segmentation result of the U-Net and your network. Even if the performances are on the same level maybe the segmentation results are showing differences.
> > > >
> > > >
> > > > Response: we believe our presentation in section 4.2 caused the confusion. In this Hippocampus experiment, we actually conducted the analysis in two stages. The first stage is basically a model selection step, where we compare the different setups for Tr-CapsNets and U-Nets, respectively, trying to select the “best” models within their respective groups. This step is carried out with only one split/fold of the data, with no intention to compare Tr-CapsNets and U-Nets. Our original presentation, which putting the results in one table, is confusing and misleading.
> > > >
> > > > We since have clarified the experiment, and separate the results into 2 tables (in the PDF), to avoid confusion.
> > > >
> > > > The second stage is about the head-to-head comparison of the best Tr-CapsNet (the 4x20 version) and U-Net (3x7). This comparisons were conducted with cross-validation, for 9 folds. In the original presentation, we only listed the average performance, where Tr-CapsNet-4x20 outperforms U-Net-3x7 (87.25 vs. 86.23). Actually, Tr-CapsNet did better than U-Net in every fold, which wasn’t mentioned in the original presentation. We have since added a new table (Table 4) to provide the details (as we cannot upload the PDF now, we include the latex of the table in the above "PDF/latex updates" comment).
> > > >
> > > > As for the improvement made by Tr-CapsNet, 87.25 over 86.23 may not seem a lot, but we believe it is quite significant and promising. Here is why. Firstly, it’s a 7.41% reduction of the error rate. Secondly, Hippocampi are difficult to segment and the ground-truth has a lot of noise. Hippocampus is a small brain structure, with very indistinct boundaries with the surrounding areas. Being small makes the Dice ratio quite sensitive to any deviation from the ground-truth. The ground-truth masks were generated by the ADNI team, in a semi-manual way. A number of 3D salient points (totally 42 for each Hippocampus) were identified by human experts, followed by a 3D surface fitting, and the surfaces were then converted to binary masks. The noise in ground-truth makes it impossible or even meaningless to shoot for 100% accuracy.  In addition, for different human raters, or even the same rater conducting the procedure twice, there will be quite amount of disparity. Actually there are studies to compare human performance, and ~87% Dice is about the disparity between experienced neurologist. Any number higher than it could be regarded as “human level intelligence”. In other words, any improvement at this level would be quite difficult. Reduction of 7.41% in the error rate should not be taken slightly.
> > > >
> > > > With all this said, this goal of this work is not to develop a most accurate Hippocampus segmentation solution. The contribution of our model lies more in the theoretical innovation, with a very sincere purpose to dig out the potential of Capsules. While we certainly agree with the reviewers that the system needs to demonstrate practical usefulness (which we certainly did), we hope we could view the significant of this work in a broader spectrum. Our trace-back pipeline, demonstrated in this work for segmentation, is a brand new idea and it would have many applications including detection, visualization and even detecting adversarials.
> > > >
> > > > There are also very recent works (in NIPS’18, PRCV’18) demonstrating capsule nets can be used for large images. With the new capsule models and ideas, our trace-back idea may lead significant stride of developments along this capsule direction.

---

> > > > > ### Author Response · Authors · 2018-12-05
> > > > > **Revisions & thoughts (part 2)**
> > > > >
> > > > > >> Finally, you trained your MNIST model with uniform noise in the ratio [1-5]. Implicitly, I assume that you apply this noise level on the real image space uint8. I thought that you applied the noise to make the segmentation more challenging. But with this small noise level, even the disturbed images are easily segmentable. Hence, I don't wanna use these results to make any judgement about your method. MNIST in this setting is to easy as segmentation task.
> > > > >
> > > > >
> > > > > Response: We agree with the review that MNIST itself is not difficult to segment. Again, the main purpose of this experience to explore the functionality of the recognition component, which we believe has been very well demonstrated with the occlusion cases.

---

### Official Review · AnonReviewer1 · 2018-11-02
**A neat idea**

**Rating:** 6
**Confidence:** 3

**Review:**

This paper proposes a traceback layer for capsule networks to do semantic segmentation. Comparing to previous works that use capsule networks for semantic segmentation, this paper makes explicit use of part-whole relationship in the capsule layers. Experiments are done on modified MNIST and Hippocampus dataset. Results demonstrate encouraging improvements over U-Net. The writing could be tremendously improved if some background of the capsule networks is included.

I have a question about the traceback layer. It seems to me that the traceback layer re-uses the learned weights c_{ij} between the primary capsules and the class capsules as guidance when “distributing” class probabilities to a spatial class probabilistic heatmap. One piece of information I feel missing is the affine transformation that happens between the primary capsule and the class capsule. The traceback layer doesn’t seem to invert such a transformation. Should it do so?

Since there have been works that use capsule networks for semantic segmentation, does it make sense to compare to them (e.g. LaLonde & Bagci, 2018) ?

---

> ### Author Response · Authors · 2018-11-24
> **Thanks & our response**
>
> >> The writing could be tremendously improved if some background of the capsule networks is included.
>
> Response: we agree with this assessment. Based this comment, we added paragraphs in the Background section (page 2) to provide more details on capsule and capsule nets. Comparisons are also made with CNNs.
>
> >> I have a question about the traceback layer. It seems to me that the traceback layer re-uses the learned weights c_{ij} between the primary capsules and the class capsules as guidance when “distributing” class probabilities to a spatial class probabilistic heatmap.
>
> Response: the assessment is accurate. We take a good use of c_ij to compute P(c_k), the class probability maps for capsule layers. We added figure 2 and rewrote the description to explain the architecture of our proposed Tr-CapsNet model, as well as the traceback procedure.
>
> >> One piece of information I feel missing is the affine transformation that happens between the primary capsule and the class capsule. The traceback layer doesn’t seem to invert such a transformation. Should it do so?
>
> Response:  In capsule net, the weight matrix W between capsule layers indeed represents a linear transformation that would map parts of an object into a cluster for the same whole. This is one of the major differences between capsule nets and CNNs. The purpose of our traceback pipeline is to estimate the (inverse) part-whole relations, based on which we can derive the class probability maps for capsule layers and later pixels. We observed such maps can be inferred through repeated applications of the product rule and the sum rule in probability theory, and only c_ij would be needed. In other words, W is not needed nor modified over the traceback pipeline. Additional introduction on capsule nets, as well as the comparisons with CNNs are added into the background section (page 2).
>
> >> Since there have been works that use capsule networks for semantic segmentation, does it make sense to compare to them (e.g. LaLonde & Bagci, 2018) ?
>
> Response: we commented on LaLonde & Bagci’s work in Discussion and Related Work section. Additional thoughts on their work and our models have been added into this revised manuscript (on page 10). We also add two paragraphs to comment on capsule based solutions in general.

---

### Official Review · AnonReviewer2 · 2018-11-03
**Original and interesting, requires further explanation of the architecture and experiment on multi-class segmentation**

**Rating:** 6
**Confidence:** 4

**Review:**

Authors present a trace-back mechanism to associate lowest level of Capsules with their respective classes. Their method effectively gets better segmentation results on the two (relatively small) datasets.

Authors explore an original idea with good quality of experiments (relatively strong baseline, proper experimental setup). They also back up their claim on advantage of classification with the horizontal redaction experiment.
The manuscript can benefit from a more clear description of the architecture used for each set of experiments. Specially how the upsampling is connected to the traceback layer.
This is an interesting idea that can probably generalize to CNNs with attention and tracing back the attention in a typical CNN as well.

Pros:
The idea behind tracing the part-whole assignments back to primary capsule layer is interesting and original. It increases the resolution significantly in compare to disregarding the connections in the encoder (up to class capsules).

The comparisons on MNIST & the Hippocampus dataset w.r.t the U-Net baseline are compelling and indicate a significant performance boost.

Cons:
Although the classification signal is counted as the advantage of this system, it is not clear how it will adopt to multi-class scenarios which is one of the major applications of segmentation (such as SUN dataset).

The assumption that convolutional capsules can have multiple parents is incorrect. In Hinton 2018, where they use convolutional Capsule layers, the normalization for each position of a capsule in layer below is done separately and each position of each capsule type has the one-parent assumption. However, since in this work only primary capsules and class capsules are used this does not concern the current experiment results in this paper.

The related work section should expand more on the SOTA segmentation techniques and the significance of this work including [2].

Question:
How is the traceback layer converted to image mask? After one gets p(c_k | i) for all primary capsules, are primary capsule pose parameters multiplied by their p(c_k |i ) and passed all to a deconv layer? Authors should specify in the manuscript the details of the upsampling layer (s) used in their architecture. It is only mentioned that deconv, dilated, bilinear interpolation are options. Which one is used in the end and how many is not clear.


Comments:
For the Hippocampus dataset, the ensemble U-Net approach used in [1] is close to your baseline and should be mentioned cited as the related work, SOTA on the dataset. Also since they use all 9 views have you considered accessing all the 9 views as well?


[1]: Hippocampus segmentation through multi-view ensemble ConvNets
Yani Chen ; Bibo Shi ; Zhewei Wang ; Pin Zhang ; Charles D. Smith ; Jundong Liu
[2]: RefineNet: Multi-Path Refinement Networks for High-Resolution Semantic Segmentation
Guosheng Lin, Anton Milan, Chunhua Shen, Ian Reid

---

> ### Author Response · Authors · 2018-11-24
> **Thanks & our response**
>
> >> The manuscript can benefit from a more clear description of the architecture used for each set of experiments. Specially how the upsampling is connected to the traceback layer.
>
> Response:​ we agree with the reviewer’s assessment. To clear up our presentation, we rewrote the description of our proposed Tr-CapsNet model, added figure 2 to illustrate the overall network structure and the traceback pipeline.
>
> >> Although the classification signal is counted as the advantage of this system, it is not clear how it will adopt to multi-class scenarios which is one of the major applications of segmentation (such as SUN dataset).
>
> Response:​ we added a paragraph titled “From capsule-based recognition to capsule-based segmentation and beyond” to explain the assumptions and limitations in the discussion section, on page 10.
>
> >> The assumption that convolutional capsules can have multiple parents is incorrect. In Hinton 2018, where they use convolutional Capsule layers, the normalization for each position of a capsule in layer below is done separately and each position of each capsule type has the one-parent assumption. However, since in this work only primary capsules and class capsules are used this does not concern the current experiment results in this paper.
>
> Response: Here is our understanding​. Just like a convolutional layer in CNN, each capsule in Layer L+1 takes capsules in an area of Layer L ( sized kernel_size * kernel_size, i.e. receptive field) as its inputs. If the stride is set smaller than the kernel_size,  “each convolutional instance of a capsule in layer L receives at most (kernel_size * kernel_size) feedback from each capsule type in layer L + 1.” (Hinton 2018). With respect to capsule networks, a significant assumption (Sabour 2017 and Hinton 2018) is that there is only one instance of the entity at a location. As for multi-instances issue, please refer to the “from capsule-based recognition to capsule-based segmentation and beyond” paragraph for our thoughts.
>
> >> The related work section should expand more on the SOTA segmentation techniques and the significance of this work including [2].
>
> Response: We did an extensive literature review on the recent FCN based semantic segmentation and included them the “Discussion and Related Work” section on page 10. On page 6, we also include a paragraph and a figure to provide more details regarding Hippocampus segmentation, as well as the SOTA solutions. We also commented and compared our work with the (sole) capsule-based segmentation solution on page 10.
>
> >> How is the traceback layer converted to image mask? After one gets p(c_k | i) for all primary capsules, are primary capsule pose parameters multiplied by their p(c_k |i ) and passed all to a deconv layer?
> Response: After P(c_k | i) is available for each primary capsule, it will be multiplied by P(i), the presence probability of the corresponding capsule, to produce P(c_k). Eqn. (1) on page 4 (revised paper) shows how P(c_k)  is generated through the traceback pipeline.
>
> >> Authors should specify in the manuscript the details of the upsampling layer (s) used in their architecture. It is only mentioned that deconv, dilated, bilinear interpolation are options. Which one is used in the end and how many is not clear.
>
> Response: we used the deconvolution scheme in Long et al. (2015) in this paper. In the revised paper, we rewrote the description to provide detailed and better presented description of the network modules.
>
> >> For the Hippocampus dataset, the ensemble U-Net approach used in [1] is close to your baseline and should be mentioned cited as the related work, SOTA on the dataset. Also since they use all 9 views have you considered accessing all the 9 views as well?
>
> Response: We cite [1] and several more SOTA solutions in the revised manuscript. The main focus of this work is to explore the power of capsules in semantic segmentation, rather than an extremely accurate hippocampus segmentation solution. The 9 views in [1] needs to combined through an ensemble net, which is not the interest of this work.

---

> > ### Comment · AnonReviewer2 · 2018-12-05
> > **Thanks for the updates**
> >
> > The added details has helped tremendously in clarifying the method and experiments. Thanks for adding the references as well. I would still suggest explaining in the paper why Chen et al is not comparable to this study.
> >
> > The only part that I'm still confused is the reason behind the averaging of convolutional probabilities: P(Ck|i) = SUM_n Pn(Ck|i)/N. A convolutional instance of a capsule routes to one location of one of the types (the parent normalization should be over [types,kernel,kernel] and not just types since as you quoted each instance receives different feedbacks for the [kernelxkernel] instances of each type in next layer. For convolutional capsules you should have P(Ck | i,h,w): what is the probability of belonging to C_k for type i at position h & w. Also the Capsule routing gives you the routing factor for both positions and types: c_{ihw}{jh'w'}.
> > Then P(Ck| i,h,w) = sum_j,k,k' c_{i,h,w}{j,h+k,w+k'} P(Ck | j, h+k, w+k'). Should I assume you are sharing the c_ij for all positions and that's why you are averaging with the same weight rather than multiplying to their routing factors?

---

> > > ### Author Response · Authors · 2018-12-06
> > > **Thanks for your inputs to improve our model**
> > >
> > > Response: Thanks for your comment. We will certainly add the explanations into the final version of the paper.
> > >
> > > The explanation of the averaging is as following:
> > > The equation P(Ck|i) = SUM_n Pn(Ck|i)/N is proposed to calculate P for a convolutional capsule in layer L (i.e. any capsule in the overlapping area of the two frames in the left of Figure 3) that receives feedbacks from capsules at more than one locations of layer L+1. Each capsule that provides a feedback is called a potential parent in the paper. For a [k*k] convolution, a capsule in layer L might have [k*k*M] potential parents in layer L+1 (M is the number of capsule types in layer L+1).
> > >
> > > Calculation of P(Ck|i) is a 2-step process. In the first step,  each Pn(Ck|i) is calculated by summing over M potential parents at the same location in layer L+1 (equation 2). Totally [k*k] such Pn(Ck|i)s would be calculated in this step. As c_ij is used to route capsules in Layer L to capsules in Layer L+1 during the forward inference, c_ij is also involved in the traceback step between the same pair. In the second step, Pn(Ck|i)s are averaged/normalized to evaluate the final P(Ck|i). Each Pn(Ck|i) is given an equal weight in this step because inference of capsules at one location is independent of capsules at other locations in the same layer. The c_ij has been included, in step 1 (Eqn. 2).

---

### Author Response · Authors · 2018-11-24
**Thank all the reviewers & overall responses**

First​ ​of​ ​all,​ ​we’d​ ​like​ ​to​ ​thank​ ​all​ ​the​ ​reviewers​ ​for​ your ​time​ ​and​ ​expertise​ ​to​ ​identify​ ​the​ ​areas​ ​of our​ ​manuscript​ ​that​ ​needs​ ​to​ ​be​ ​corrected​, clarified and improved.​ ​We have carefully read through all your comments and implemented additional components to thoroughly address your concerns, which will be explained in the following paragraphs.

We’ve integrated many updates into the revised manuscript. To assist you to navigate through them, we summarize the major changes as follows:

1) All the sections been updated. The added & rewritten sentences are highlighted with red color.
2) We renamed the traceback layer to traceback pipeline, to better reflect the nature of the operations.
3) Background section: added paragraphs to provide more details on capsule and capsule nets. Comparisons are also made with CNNs.
4) Architecture section:
       4.1) rewrote the description of our proposed Tr-CapsNet model, with an additional figure to illustrate the overall network
           structure, as well as the traceback pipeline.
       4. 2) updated the text and caption of Figure 3,
5) Experimental results section:
       5.1) additional, more detailed description on the Hippocampus dataset.
       5.2) provide more implementation details.
       5.3) implemented and compared different trackback depths.
6) Discussion and related work section:
       6.1) conducted additional literature review on state-of-the-art solutions.
       6.2) discussion on CapsNets’ single-instance assumption, and its influence to the capacity of our model. Proposed
       6.3) directions for future enhancements.
7) Added an Appendix to provide the detailed network configurations.

---

> ### Author Response · Authors · 2018-12-05
> **new updates**
>
> We revised the manuscript (pdf) with the following additions. However, the PDF cannot be uploaded again, so we mention the items here, and include the latex in next comment.
>
> 8) added an illustration of the segmentation results in Appendix B. The ground-truth masks (green color), results from Tr-CapsNet (left column, red) and U-Net (right column, magenta) of two slices are included.
>
> 9) Updated section 4.2: results on Hippocampus dataset. A table (Table 4) showing the segmentation accuracies of the best Tr-CapsNet and U-Net in all the folds is added.

---

> > ### Author Response · Authors · 2018-12-05
> > **PDF/latex updates**
> >
> > The new updates (Dec. 5) we made to the latex/pdf file include the following:
> >
> > 1) we separate Table 3 in the original pdf into two tables, for model selection of Tr-CapsNets and U-Nets respectively.
> >
> > 2)  The best Tr-CapsNet model (4x20) actually outperforms the best U-Net (3x7) in every fold of the final cross-evaluation.  We therefore added a table (Table 4) to include the numbers, and updated the "Overall average performance" paragraph in section 4.2, as follows:
> >
> > {\bf Overall average performance} Through the one-split validation, we identified the potentially best setups for both Tr-CapsNet and modified U-Net, which are Tr-CapsNet- 4×20 and U-Net-3×7, respectively (refer to APPENDIX A for detailed layer configurations). We then carried out a nine- fold cross-validation on the entire 108 data points. Table 4 shows the segmentation accuracies of Tr-CapsNet and U-Net in all the nine folds, followed by the averages. As evident, Tr-CapsNet obtain a higher Dice ratio in each fold. On average, the Dice ratio for Tr-CapsNet- 4×20 is 87.25 with a standard deviation 5.05, while U-Net-3×7 obtains 86.23 with a standard deviation 2.19. In other words, our model outperforms the best U-Net model.
> >
> > \begin{wrapfigure}{R}{0.45\textwidth}
> > \begin{tabu}{c c c  }
> >     \hline
> >     \hline
> >     Fold & Tr-CapsNet & U-Net  \\
> >     \hline
> >     1 &   \makecell{ \textbf{88.86} $\pm$ 1.628}  &   \makecell{88.41 $\pm$ 1.707 }  \\
> >     2 &   \makecell{ \textbf{87.84} $\pm$ 5.674} &  \makecell{86.03 $\pm$ 3.347 } \\
> >     3 &   \makecell{ \textbf{86.53} $\pm$4.678} & \makecell{ 85.42 $\pm$ 3.005 } \\
> >     4 &   \makecell{\textbf{85.73}$\pm$ 7.352}  & \makecell{85.21 $\pm$ 2.301 } \\
> >     5 &   \makecell{\textbf{87.09}$\pm$ 4.190} &  \makecell{86.53 $\pm$ 2.159 }  \\
> >     6 &   \makecell{\textbf{83.56} $\pm$ 1.257} &  \makecell{82.34$\pm$ 1.369}  \\
> >     7 &   \makecell{\textbf{88.57} $\pm$ 1.783} &  \makecell{ 87.14 $\pm$ 1.374 } \\
> >     8 &    \makecell{\textbf{88.23}$\pm$ 4.506} &  \makecell{86.96 $\pm$ 2.071 } \\
> >     9 &   \makecell{\textbf{88.82}$\pm$ 3.005} & \makecell{88.03 $\pm$ 2.333 }  \\
> >     \hline
> >     \textcolor{red}{Average} & \makecell{\textbf{87.25} $\pm$ 3.786} & \makecell{86.23 $\pm$ 2.193 } \\
> >     \hline
> > \end{tabu}
> > \captionof{table}{\textcolor{red}{Hippocampus segmentation accuracies on nine-fold
> >     cross-validaton.}}
> > \label{tab:hippo}
> > \end{wrapfigure}

---

### Meta-Review · Area_Chair1 · 2018-12-10
**metareview: insufficient experimental validation**

**Confidence:** 5
**Recommendation:** Reject

**Metareview:**

This paper proposes a method for tracing activations in a capsule-based network in order to obtain semantic segmentation from classification predictions.

Reviewers 1 and 2 rate the paper as marginally above threshold, while Reviewer 3 rates it as marginally below. Reviewer 3 particularly points to experimental validation as a major weakness, stating: "not sure if the method will generalize well beyond MNIST", "I’m concerned that the results are not transferable to other datasets and that the method shines promising just because of the simple datasets only."

The AC shares these concerns and does not believe the current experimental validation is sufficient. MNIST is a toy dataset, and may have been appropriate for introducing capsules as a new concept, but it is simply not difficult enough to serve as a quantitative benchmark to distinguish capsule performance from U-Net. U-Net and Tr-CapsNet appear to have similar performance on both MNIST and the hippocampus dataset; the relatively small advantage to Tr-CapsNet is not convincing.

Furthermore, as Reviewer 1 suggests, it would seem appropriate to include experimental comparison to other capsule-based segmentation approaches (e.g. LaLonde and Bagci, Capsules for Object Segmentation, 2018). This related work is mentioned, but not used as an experimental baseline.